# Smart Grid Security: An Effective Hybrid CNN-Based Approach for Detecting Energy Theft Using Consumption Patterns

**DOI:** 10.3390/s24041148

**Published:** 2024-02-09

**Authors:** Muhammed Zekeriya Gunduz, Resul Das

**Affiliations:** 1Department of Computer Science and Technology, Vocational School of Technical Sciences, Bingöl University, Bingöl 12000, Türkiye; 2Department of Software Engineering, Technology Faculty, Firat University, Elazığ 23119, Türkiye; rdas@firat.edu.tr

**Keywords:** convolutional neural network, cyber security, deep learning, energy theft, generative adversarial network, Internet of Things, smart grid

## Abstract

In Internet of Things-based smart grids, smart meters record and report a massive number of power consumption data at certain intervals to the data center of the utility for load monitoring and energy management. Energy theft is a big problem for smart meters and causes non-technical losses. Energy theft attacks can be launched by malicious consumers by compromising the smart meters to report manipulated consumption data for less billing. It is a global issue causing technical and financial damage to governments and operators. Deep learning-based techniques can effectively identify consumers involved in energy theft through power consumption data. In this study, a hybrid convolutional neural network (CNN)-based energy-theft-detection system is proposed to detect data-tampering cyber-attack vectors. CNN is a commonly employed method that automates the extraction of features and the classification process. We employed CNN for feature extraction and traditional machine learning algorithms for classification. In this work, honest data were obtained from a real dataset. Six attack vectors causing data tampering were utilized. Tampered data were synthetically generated through these attack vectors. Six separate datasets were created for each attack vector to design a specialized detector tailored for that specific attack. Additionally, a dataset containing all attack vectors was also generated for the purpose of designing a general detector. Furthermore, the imbalanced dataset problem was addressed through the application of the generative adversarial network (GAN) method. GAN was chosen due to its ability to generate new data closely resembling real data, and its application in this field has not been extensively explored. The data generated with GAN ensured better training for the hybrid CNN-based detector on honest and malicious consumption patterns. Finally, the results indicate that the proposed general detector could classify both honest and malicious users with satisfactory accuracy.

## 1. Introduction

The development of the Internet has enabled more effective and widespread use of Internet of Things (IoT) applications. IoT enables the connection of different objects to the Internet and the ability to communicate with devices in distant networks [1]. Critical infrastructures such as electricity grids have become IoT-based [2]. Electricity generation, transmission, distribution, and consumption processes have become more manageable in this way. IoT-based electricity systems are called the smart grid. The advanced metering infrastructure (AMI) is the communication network of smart grid applications [3]. The AMI carries sensitive information, making it a potential target for attackers. Due to the inherent vulnerabilities of communication networks, cyber-security emerges as a leading problem in smart grid systems [4].

The daily life of humankind depends on electricity and requires effective management. AMI helps this management by using control commands and real-time transmission of the data to utilities, customers, and third parties. Generally, an AMI system consists of smart meters, gateways, communication networks, and a headend system [5]. The most prominent component of AMI is smart meters. Smart meters increase the frequency of collection of energy consumption data, enabling advanced data analysis that was not possible before [6]. A smart meter records and transmits energy consumption of the customers at specific intervals for billing and management [7]. Unauthorized access to a smart meter may result in data tampering attacks called energy theft [8]. Energy theft is a significant challenge for smart grid applications as malicious actors continue to exploit potential vulnerabilities [9]. Unethical customers represent the highest probability of threats to the AMI and smart meters. In the past, energy theft mainly involved physical disruptions like cut-offs or damage. However, contemporary instances may encompass sophisticated attacker models, including erasing log events, false data injection (FDI) attacks, intercepting communication, and data manipulation [10].

Energy theft is a significant concern for utilities, and it has emerged as a global issue, resulting in technical and economic losses for operators and governments [11]. Deep learning (DL)-based models play a prominent role in the design of effective intrusion detection systems (IDSs). Such IDSs are used to identify abnormal activities such as FDI and data tampering [12]. Energy theft is an important issue that needs to be solved to improve smart grid applications. Also, information and communication technologies (ICTs) and correlated cyber-threats necessitate proactive measures. There are various studies on energy theft detection handling the consumption data to achieve a high detection rate (DR) and accurate results [13,14,15,16,17]. Many methods are used for energy theft detection, such as statistics, data mining, machine learning (ML), and DL techniques [18]. DL-based IDSs play a critical role in identifying energy theft attacks [19].

We focus on further investigating CNN-based architecture for energy theft detection within real smart meter consumption data. We also discuss balanced dataset generation, which is important to increase performance in DL with GAN. We proposed a CNN-based deterministic model to detect energy theft based on consumption patterns. The symptom state parameters were used to assess the likelihood of energy theft incidents. Our model monitors energy consumption patterns and performs with high accuracy in detecting maliciously changed data in experimental results. Also, it can learn normal/abnormal behaviors obtained from the consumption data of the smart meters and detect anomalies based on deviation from the probability.

Unpredictable attack vectors may be considered zero-day attacks [15]. We address the challenges posed by zero-day attacks and imbalanced data by generating synthetic attack datasets, leveraging the predictable nature of theft patterns. Extensive trials demonstrate a notable enhancement in the DR and the capability to detect various types of attacks. In addition to detecting a thief, the proposed IDSs can also precisely identify the time of the theft.

In this study, we employ a hybrid structure by combining CNN with shallow ML techniques to develop an energy theft detector. This hybrid approach is applied to the energy consumption dataset, addressing the classification problem and allowing us to exploit irregular and abnormal consumption patterns for energy theft detection. The hybrid structure includes CNN combined with Support Vector Machine (SVM), Random Forest (RF), Decision Tree (DT), k-Nearest Neighbors (KNN), and Logistic Regression (LR). CNN is used for feature extraction, while the other techniques are employed for classification.

### 1.1. Research Contributions

The main contributions of this paper can be summarized as follows:Data-driven energy theft detection solutions are reviewed for comparison. The results verify the effectiveness of the proposed approach.An integrated CNN-based architecture is used to tackle miss-classification and high False Positive Rate (FPR) issues.Real honest data are exposed to six attack vectors to create theft classes.GAN technique is used to eliminate the imbalanced data issue.To overcome the over-fitting problem, k-fold cross-validation technique is opted for.Six balanced datasets containing the characteristics of only one attack vector were created. Additionally, a single dataset combining these six attack vectors was designed. Thus, both attack-vector-specific solutions and a general solution have been proposed.The performance of the hybrid CNN-based model was evaluated by comparing it with other up-to-date approaches.

### 1.2. Organization of the Paper

This study emphasizes the importance of smart grid security, non-technical loss (NTL), the design of a CNN-based energy theft detector, and the generation of a balanced dataset. The structure of the remaining paper is as follows. Related works in the relevant literature are presented in Section 2. Next, Section 3 presents the dataset, attack vectors, and GAN methodology, along with a CNN-based hybrid approach. Then, Section 4 deals with the experimentation and the results of the proposed model, providing detailed insights into their outcomes and conducting a comparative analysis with other algorithms and studies. Finally, Section 5 provides a conclusion after performing the experiments in this paper and presents future work.

## 2. Related Works

Many vulnerabilities inherited from communication networks exist in AMI. The basic requirements of cyber-security are availability, integrity, and confidentiality [20]. In this study, we have examined data tampering attacks specifically designed to undermine the integrity of consumption data. We aimed to detect attack vectors that reduce smart meter readings. DL-based IDSs have provided high accuracy in detecting these attack vectors in the literature [21].

Understanding the data flow in smart grid applications is significant, and this can be achieved by examining their general structure. The overall structure of the smart grid environment is shown in Figure 1. Energy generated from diverse sources is transmitted over long distances through transmission lines and distributed to consumers via distribution lines. Data transmission is provided through AMI in the context of the energy infrastructure. While the Wide Area Network (WAN) is used in generation and transmission domains, the Neighborhood Area Network (NAN) and Field Area Network (FAN) are used in the distribution domain. Lastly, the Home Area Network (HAN) and Industrial Area Network (IAN) are used in the consumption domain.

Energy theft detection in smart grids has been an active research area in recent years. The literature has introduced various strategies for detecting energy theft. These strategies include state estimation, game theory, and data-driven strategies. Data-driven strategies [22] are more prevalent due to their scalability for handling large systems and their cost-effectiveness in computational resources. Statistics, data mining, ML, and DL are among the prominent data-driven methods extensively employed to extract knowledge from consumption patterns, enabling inferential assessments. While detecting NTLs involves challenges, smart meters allow the extensive storage of energy data, enabling various analytical approaches. This has led to the development of various classification techniques.

In this section, we critically reviewed notable studies on energy theft detection that utilize smart meter consumption data to identify potential attack vectors and malicious customers. Due to their higher performance, we focused on the ML and DL approaches.

Jokar et al. [23] propose an energy theft detector within AMI based on consumption patterns, utilizing the SVM approach. The detector enhances the classification accuracy to 94%. Moreover, it addresses a range of cyber-attack vectors associated with energy theft, and these are widely acknowledged in the literature. The authors of [24] introduced a two-step energy-theft-detection system utilizing DT and SVM, achieving an accuracy of 92.5%. However, there is no information on whether the dataset is balanced or imbalanced. The researchers in [25] present an energy-theft-detection method utilizing ensemble ML models. The concept behind the models involves combining various ML methodologies into a unified predictive model to increase DR and decrease the error rate. The results indicate that a bagging-type ensemble ML approach, which aggregates the outcomes of independent ML models in parallel through averaging, outperforms a boosting approach. However, when compared to other approaches, the recommended model has not demonstrated better success.

Despite the absence of a real dataset in [26], notable achievements in performance were attained through the application of a neural network. They achieved an overall DR of 93%. The authors of [27] have devised a novel approach for identifying and detecting energy theft within distribution systems, employing the multilayer perceptron artificial neural network (MP- ANN). They achieved a successful differentiation between malicious and honest users, averaging a detection rate of 93.4%. However, there is no information on whether the dataset is balanced or imbalanced. In [28], a hybrid deep neural network (DNN) approach is proposed. The gated recurrent unit (GRU) technique was used, which is an evolved variant of LSTM belonging to the category of recurrent neural networks (RNNs). The hybrid DNN combines CNN, GRU, and particle swarm optimization (PSO). However, when compared to other approaches, the recommended hybrid model has not quite demonstrated better accuracy, and the proposed model tends to overfit. The work referenced as [29] employed a deep RNN classifier using GRU to catch temporal correlations within individual customer load profiles, thereby introducing a detector with a DR reaching up to 93%. However, it is not clear whether the dataset is balanced or imbalanced. In [30], the authors present a CNN model to detect energy theft, utilizing the State Grid Corporation of China (SGCC) dataset. They illustrate energy consumption over four weeks for randomly selected honest and malicious consumers. Initially, consumption is displayed by dates and later by weeks. Date-based representation fails to differentiate between honest users and thieves, but the weekly representation distinguishes them. Honest consumers show periodic energy usage, while the thieves display less periodicity. However, there is no information on whether the dataset is balanced or imbalanced. The researchers in [31] presented a hybrid model on energy consumption patterns to detect energy theft with CNN and long short-term memory (LSTM), using the SGCC dataset. The CNN autonomously identified and categorized features, whereas the LSTM managed the sequential nature of the time-based data. The authors solved the imbalanced dataset problem by applying the synthetic minority over-sampling technique (SMOTE) method to augment the NTL class, equalizing it with honest customer counts. While achieving an 89% accuracy, the model demonstrated a lower DR of nearly 87%. Compared to other approaches, the recommended hybrid model has not demonstrated better accuracy. Adil et al. [32] used the CNN-LSTM approach on the SGCC dataset and achieved 87.9% accuracy. However, compared to other approaches, the proposed model is not very satisfactory. Kocaman and Tümen [33] introduced an LSTM classifier for identifying malicious customers. They utilize data selection, normalization, and weight updating as preprocessing steps. The LSTM classifier architecture comprises LSTM cells, dropout layers, ReLu activation functions, and a softmax classifier. Evaluation involves precision, accuracy, and recall metrics for assessing model performance. However, it is unclear how they resolved the issue of the imbalanced dataset.

The authors in [34] used the Irish Social Science Data Archive (ISSDA) dataset. They employed cluster-based algorithms, specifically the fuzzy Gustafson–Kessel and fuzzy c-means, achieving a 74.1% area under the curve (AUC). However, they achieved low true positive rate (TPR) and high FPR, which are 63.6% and 24.3%, respectively. Lastly, the authors of [35] describe an energy-theft-detection method using data about power provider system consumption at the edge. Centralized data centers employ K-means clustering and DNN to extract features. CNN refines daily, weekly, and monthly patterns. RF at the edge data center classifies the characteristics, speeding up the edge computing processing. This approach is more accurate and computationally efficient than previous methods, making it suitable for edge data centers.

Approaches using only traditional ML models often face challenges in extracting distinct consumption patterns due to the complex structure of power consumption data. This situation leads to low performance and accuracy. On the other hand, DL models can better explore complex structures, thus achieving higher success than ML models. Table 1 summarizes prominent ML- and DL-based approaches for developing energy theft detectors.

Glancing at these noteworthy works, we studied novel CNN-based hybrid models for energy theft detection and proposed a CNN-based deterministic model to detect energy theft based on consumption patterns. CNN automatically captures the distinct features of consumption behaviors from the data. It is very important for the effectiveness of energy-theft-detection models. We conducted a comparative analysis using ML and sigmoid classifiers to detect consumption patterns based on extracted features, aiming to enhance detection performance. Hybrid solutions using both CNN and traditional ML methods have been observed to achieve higher TPR and lower FPR compared to pure DL solutions. Our hybrid structure achieves a higher accuracy performance rate of up to 96% compared to existing models. On the other hand, the data-balancing process is an inherently hard mission, and working with imbalanced data typically reduces the performance of the model. While the literature generally works on imbalanced datasets, our study addresses this challenge using GAN.

## 3. Materials and Methods

Consumption data obtained from the ISSDA dataset were subjected to data analysis and preprocessing. Malicious synthetic data were generated by exposing the real data to the attack vectors. Real and malicious data were aggregated, and the prepared datasets were used to train the proposed models. In this section, the ISSDA dataset, data analysis and preprocessing processes, attack vectors, and recommended hybrid CNN-based approaches are presented. Additionally, the GAN method used in generating synthetic data is presented.

### 3.1. Dataset

The dataset, released by ISSDA [44] in January 2012, comprises electricity consumption data from over 5000 Irish residential and enterprise consumers during 2009 and 2010 in kWh for 536 days. A vector of 48 readings was used to explain the everyday usage of a consumer. This dataset, presenting a lot of samples from diverse consumers, serves as an excellent resource for research in smart meter data analysis. The dataset contains only honest profiles and reports the consumed real power data at a rate of 30 min. The daily load profile of a customer is $x=\{x_{1},\ldots ,x_{48}\}$. The dataset is scalar.

One of the challenges faced by this research is the absence of malicious data needed to train the models. Applying the attack vectors described in the next section on the real dataset solved this problem. We obtained balanced datasets to ensure the accuracy of the proposed models. After data analysis, 2104 samples were selected from the ISSDA dataset.

The class distribution in a dataset associated with a classification problem should be balanced; otherwise, it leads to an imbalanced data problem. When classes exhibit significant numerical differences, it constitutes an imbalanced data issue. The issue prevents the effective training of ML models. Furthermore, it can adversely affect the generalizability of the model and performance.

The dataset denoted as *H* consists of 2104 samples representing honest data. When the *H* dataset was exposed to the $f_{1}$ attack, a final dataset was created containing 4208 samples. The final dataset includes a balanced number of both honest and malicious samples. The same scenario holds for the other five attack vectors. These six different datasets were used only in training the proposed approaches to detect the relevant attack vector. Furthermore, we aimed to design a more comprehensive IDS. So we generated a dataset containing all attack vectors along with the *H* dataset. The dataset is structured as $H+f_{1}+f_{2}+f_{3}+f_{4}+f_{5}+f_{6}$. As observed, there is an imbalanced data problem since the number of *H* samples is 2104 and the number of attack vector samples is 6 × 2104. The imbalanced data problem was solved using the GAN approach. After that, we call the obtained dataset as the *“hybrid dataset”*. To the best of our knowledge, there are limited studies that solve imbalanced data problems with GAN in the energy-theft-detection field [45].

Data analysis and data preprocessing are some of the most important steps that affect the accuracy of the system. Data preprocessing refers to transforming the initial data into a standard format. It has three major phases: data cleaning, data standardization, and feature engineering. Data cleaning is the discovery of faults or null values in the dataset. Data standardization is the detection of outliers in the dataset and transforming them into acceptable types using scaling ways. We do not use data standardization because our consumption data do not have outliers. The feature engineering process is used to extract major features [46]. Since our dataset has limited attributes (SMID, readings (kWh), and output), there is no need for a feature engineering process.

*Data Analysis and Cleaning:* To design a general IDS capable of detecting malicious activities among residential consumers, a specific range of consumption values was chosen for the samples. Industrial users and small/medium-sized enterprises were eliminated to ensure a more normal distribution. So consumers were filtered as residential users. Consumers with missing data were not used as samples. To prevent over-fitting and under-fitting, users whose consumption data are zero for a very long time are eliminated. Therefore, every consumption data point in 30 min is chosen between 0.001 kWh and 7 kWh. Consumers with a total consumption data between 536 kWh and 12,000 kWh for 536 days were selected as the samples. Additionally, consumers with missing data in the 536 days were not selected. As a result, data from 2104 customers were analyzed. A matrix of size 536 × 48 was created for each customer. The input data were given to the CNN network. Therefore, the choice of input data form in image processing [47] contributed to achieving high accuracy.

### 3.2. Attack Vectors

The implementation of smart meters and the integration of a cyber layer into the metering system have introduced novel attack vectors for energy theft. The main purpose of attack vectors is to reduce the bill. Theft samples rarely exist or do not exist for customers. Therefore, obtaining a real dataset with malicious samples to design an energy theft detector is a challenge.

Along with their versions, the six attack vectors proposed by Jokar et al. are remarkably well accepted in the literature [23]. We applied six attack vectors on honest consumption data. The formulation of attack vectors is given in Table 2. Each honest half-hourly reading is represented as $x_{t}$. *X* represents all 536 days of a consumer’s readings as 1D and is presented as $X=\{x_{1},x_{2},\ldots ,x_{\left(536\times 48\right)}\}$. Attack vectors are represented as f(x). α and λ are randomly generated numbers between 0.1 and 0.8. In the literature, a common approach to prevent consumption data from dropping to zero and remaining unchanged is by generating random values for parameters α and λ within the specified range of 0.1 to 0.8 [23].

The $f_{1}$ attack multiplies all consumption readings by the same randomly chosen value α. The $f_{2}$ attack assigns zero to certain time interval readings of the day, and the others remain the same actual value. We chose the time period between 38 and 43 due to the peak electricity consumption during the evening hours. The $f_{3}$ attack multiplies each meter reading by a distinct random value λ. The $f_{4}$ attack generates a random number between 0.1 and the max consumption value of each user. It subtracts this result from each reading for that customer. If the result value is negative, it is set to 0; if it is positive, the difference is written. The $f_{5}$ attack subtracts the mean of all consumption data of each consumer from each reading. If the result is greater than or equal to 0, the difference is written; otherwise, the consumption data remain the same. The $f_{6}$ attack reverses the order of each customer’s readings daily. Although the total consumption remains constant, the reporting of intensive usage is shifted to low-tariff periods. The impact of the six attack vectors on the daily use of an honest customer is shown in Figure 2.

Our CNN-based approach is designed to detect anomalies within consumption patterns. Considering that every attack leads to the manipulation of meter readings, our CNN-based approach effectively detects all presented attack vectors. Furthermore, we solved the imbalanced data problem for *hybrid dataset* using the GAN approach described below.

Since there were fewer real samples in the mentioned honest dataset, we used GAN to produce honest samples. A GAN consists of a generator (G) neural network and a discriminator (D) neural network. The aim is for the generator to create synthetic data that are indistinguishable from real data, while the discriminator aims to correctly differentiate between real and synthetic data [48].

The *G* component maps a latent noise vector *z* to a synthetic honest sample. G aims to produce synthetic honest samples. G(z) represents the generated synthetic honest sample.

The *D* component distinguishes between real honest samples and synthetic ones. D(x) represents the probability that *x* is a real honest sample. D(G(z)) represents the probability that the generated instance G(z) is a real honest sample.

The objective function for the generator *G* and discriminator *D* in the context of honest sample generation can be represented as
$$\begin{array}{cc} \min_{G}\min_{D}V\left(D,G\right) & =E_{x∼p_{honest}\left(x\right)}\left[\log D(x)\right]\\ \; & +E_{z∼p_{z}\left(z\right)}\left[\log\left(1-D\left(G(z)\right)\right)\right]. \end{array}$$

Here, $p_{honest}\left(x\right)$ is the distribution of real honest samples. $p_{z}\left(z\right)$ is the prior noise distribution of *z*. E represents the expected value. *x* represents a real honest sample. *z* represents a noise vector. G(z) generates a synthetic honest sample. D(x) is the output of the discriminator for real honest samples. D(G(z)) is the output of discriminator for synthetic honest samples.

Similar to the standard GAN setup, the objective is for the generator *G* to minimize the probability of the discriminator *D* to correctly distinguish between real and synthetic honest samples, while the discriminator aims to maximize this probability.

By training the GAN iteratively, the generator learns to produce synthetic honest samples that capture the characteristics of honest data, contributing to the mitigation of imbalanced datasets in energy theft detection.

### 3.3. The Novel CNN-Based Hybrid Approach

Confidentiality, integrity, and availability are the key elements of information security. Energy theft attacks target the integrity of data. In this section, our goal is to create a top-tier classifier capable of identifying cyber-attacks aimed at compromising the integrity of the readings. The design of this detector relies on CNN, enabling the capture of complex patterns within the data. We utilized the empirical method for the selection of parameters. The SVM, RF, DT, KNN and LR algorithms were used with CNN for classification.

We tested the performance of the proposed model on real data of 2104 customers against six attack vectors of energy theft. We have compared each attack vector and obtained the most successful hybrid model for that attack. This paper presents a robust model for NTL detection in a smart grid using a CNN-based approach.

The architecture of the proposed CNN-based model is shown in Figure 3. The main structure of our CNN model is given as follows:

Convolution layers are used to learn feature representation of data. Also, convolution filters provide noise reduction. Filters are used to reduce network size. Thus, lower computational complexity is provided. Our CNN model is designed using an input layer, 3 convolution layers, 3 max-pooling layers, 2 dropout layers, 2 fully connected layers, and an output layer. The architecture of hybrid CNN-based model accepts consumption data with a size of 536 by 48 in the input layer. The ReLU activation function and L2 regularization layers were preferred to prevent over-fitting in the convolution layer. There are max-pooling layers from each convolution layer. After the max-pooling process, dropout is applied. Feature maps in the first fully connected layer were classified with ML algorithms. In addition, feature vectors for the pure CNN model were classified by giving them to the sigmoid activation function in the output layer.

The relevant equations of the operations are explained below. Convolution blocks are connected to the max-pooling layers. The mathematical output of the convolution layers is stated as
$$y_{conv}\left(X_{l}^{f_{l}}\right)=\delta \left(\sum_{f_{l}=1}^{F_{l}}W_{l}^{f_{l}}*X_{l}^{f_{l}}+b_{l}^{f_{l}}\right),$$
where δ and ∗ show the activation function and convolution operation, respectively. $b_{l}^{f_{l}}$ and $W_{l}^{f_{l}}$ show learnable parameters in the f-th feature filter. Dropout layers are located between two convolution layers. These layers provide dimensional reduction by reducing the number of parameters. Generally, the min-pooling and max-pooling approaches are used for this aim. Our CNN model uses max-pooling. This operation generates more efficient results. The mathematical definition of this operation is expressed as
$$y_{pool}\left(X_{l}^{f_{l}}\right)=\max_{m\in M}\left\{X_{l,m}\right\},$$
where *M* and *m* indicate a set of activations and the index of activations in the pooling window, respectively. After the dropout layer, a fully connected layer is applied for flattening feature maps into one feature vector as follows:$$y_{f_{l}}\left(X_{l}\right)=\delta \left(W_{l}\;.\;X_{l}+b_{l}\right),$$
where $W_{l}$ and $b_{l}$ show the weight and bias of the l-th layer. We used the sigmoid classifier for the output layer in the CNN. An example is a CNN that uses the RF classifier defined as
$$y_{out}\left(X_{L}\right)=sigm\left(W_{rf}\;.\;X_{l}+b_{l}\right),$$
where sigm(.) is the sigmoid function. It maps malicious values to 0 and honest values to 1.

A CNN-based IDS can identify data that deviate from expected patterns, assuming that the patterns of energy theft differ from those of honest users. Achieving high accuracy and DR in anomaly detection models includes selecting proper activation functions or optimizing parameters.

Hybrid models are significant approaches for achieving high accuracy and DR in energy theft detection. In our study, we focused on individually detecting each attack vector. Additionally, we achieved high DR of all six attack vectors when they were collectively present in the *hybrid dataset*. We show these different situations in two flow charts. The overall flowchart is presented for individual attack vectors in Figure 4. Moreover, the overall flowchart is presented for the combined set of all attack vectors in Figure 5.

Figure 4 shows the NTL flowchart of the proposed model for each attack vector. Firstly, the dataset obtained from ISSDA was subjected to preprocessing by performing data analysis and cleaning. The obtained honest samples were exposed to each *f* attack vector separately, and six balanced datasets were obtained. Five CNN-based ML models and a pure CNN model were employed to classify users as honest or malicious. Each CNN-based model was trained and tested on these balanced datasets. Finally, the performance of the models was compared.

Figure 5 shows the NTL energy theft detection for all attacks collectively. Firstly, datasets for the six attack vectors were generated from the honest dataset. Afterward, the honest dataset was augmented nearly sixfold using GAN to solve the imbalanced data problem. Then, the honest dataset and the datasets containing *f* attacks were combined. A five-fold cross-validation technique is used to prevent over-fitting. The obtained dataset samples were classified as honest or malicious by CNN-based classifiers. Model performances were evaluated using the confusion matrix and receiver operating characteristic (ROC) curves. Finally, the models were compared.

As a result, the materials and methods employed in this study serve as the foundation for our experimental framework, ensuring rigorous data collection, preprocessing, and model implementation. The careful selection of datasets, the application of advanced algorithms, and the systematic validation processes collectively supported the credibility and reliability of our results. The utilization of state-of-the-art methods and the adherence to standardized protocols underscored the robustness of our experimental setup, facilitating a comprehensive analysis of the research objectives.

## 4. Results

Some of the experimental studies were conducted using an A5000 with 24 GB GPU and 128 GB RAM. Additionally, the data preprocessing and training processes utilized a V100 GPU with 50 GB RAM through Google Colab Pro. The Python 3.10 programming language and the TensorFlow and Scikit-learn libraries were employed. The hyperparameters for the designed models were set as follows: epoch value 100, batch size value 64, optimizer SGD, learning rate 0.01, L2 regularization rate 0.001, and binary cross-entropy as the loss function.

### 4.1. Training and Testing

The dataset was subjected to certain analysis and preprocessing. As a result of the processing, the generated datasets contain the electricity consumption data of 2104 honest and 2104 malicious customers within 536 days with half-hour intervals. The datasets were divided using a five-fold cross-validation technique into training and testing sets. Malicious ones were generated synthetically. It is demanding to extract features based on experience from the 1D electricity consumption data since daily consumption fluctuates in a relatively independent way. Thus, we designed a CNN solution to process the electricity consumption data in a 2D manner. We defined 2D data as a matrix of actual energy consumption values for a specific customer, where the rows of the matrix represent days as D={1,…,536} and columns represent the time periods as T={1,…,48}. The feature maps from the fully connected layer were utilized as inputs for ML algorithms. Furthermore, classification was carried out using the sigmoid function in the output layer of the CNN.

Consumption habits of customers can vary due to several non-malicious factors. These variations can be temporary, periodic, or permanent, potentially leading to false positive outcomes. Short-term changes may result from unusual behaviors, such as hosting an event that lasts more than a day or two. Such examples are included in training and testing datasets to mitigate this issue.

A DL-based architecture requires large amounts of data. In this study, datasets were generated with large amounts of data in order to train the proposed CNN architecture correctly. The number of samples in the datasets is presented in Table 3.

### 4.2. Evaluation Metrics

In this section, the metrics used to evaluate the performance of the proposed energy-theft-detection models are mentioned. The confusion matrix was used in the performance analysis of the developed models. The confusion matrix is a prominent structure for addressing the classification performances of the models [49].

The confusion matrix divides the entire dataset into True Positive (TP), False Positive (FP), False Negative (FN), and True Negative (TN). TP represents positives correctly predicted as positives. FP represents negatives incorrectly predicted as positives. FN represents positives incorrectly predicted as negatives. TN represents negatives correctly predicted as negatives [50]. The structure of the confusion matrix is presented in Figure 6.

The performance of the proposed approach on six different classifiers was tested using the accuracy, precision, specificity, TPR, FPR, and F1-score. Moreover, the Highest Difference (HD) is defined as the difference between the TPR and FPR. Additionally, Recall, sensitivity, TPR, and DR refer to the same metric. Also, FPR and False Alarm (FA) generally refer to the same metric in these studies. Since all of these concepts are used in these studies, we have presented them to avoid conceptual confusion. The metrics defined to evaluate the performance of the models are presented in Table 4.

The ROC curve is an another significant performance metric for evaluating binary classification models. ROC curves depict the trade-off between TPR and FPR. As the discrimination threshold of a binary classifier varies, the TPR and FPR change consistently. The track of (FPR, TPR) is a curve connecting (0, 0) and (1, 1). The AUC metric is used in evaluating classification models by measuring the area under the ROC curve. A higher AUC indicates better overall performance. The AUC metric is commonly used to compare different models or to select the best-performing model among several candidates for a particular classification task.

Each metric offers insights into different aspects of a model’s performance and helps in understanding its strengths and weaknesses in making predictions in binary classification tasks. Accuracy gives an overall view. Focusing on accuracy, precision, recall, F1-score, and AUC can offer a comprehensive understanding of the model’s effectiveness in balanced datasets. When dealing with imbalanced datasets precision, recall, F1-score, and AUC offer a better insight. It was observed that performance metrics were chosen differently in many studies examined within the scope of this study. In general, it has been observed that the overall performance of the model is better when accuracy, DR, and HD are high and FPR is low.

### 4.3. Evaluation of the CNN-Based Models

The overall detection performances were tested for each attack vector, as well as for the *hybrid dataset* encompassing various attack vectors. Evaluation metrics were presented for each algorithm. The performance of algorithms varied across different attack scenarios and balanced datasets. We used five-fold cross validation and presented the mean of the folds with standard deviation in the tables. SVM, LR, RF, KNN, DT algorithms were used with CNN for classification. In order to determine the best CNN-based model, a comparative analysis was performed. Since our hybrid models work on balanced datasets, the accuracy, DR, F1-score metrics stand out in terms of evaluating the prominent performance metrics.

The performance metrics of the models and datasets with a single attack vector are given in Table 5. While the all models were generally effective in detecting all attack vectors, they performed poorly for #5. It was found that #2, #3, #4, #5 and #6 were detected with 99.81%, 98.04%, 99.48%, 81.42%, and 98.53% DR, respectively, using a CNN+RF model, and #1 was detected with a DR of 96.14% by CNN+LR. The CNN+RF model nearly obtained the best accuracy up to 99.83% except for #1. It closely followed by the CNN+LR up to 99.67%. The standard deviation values of the folds are generally close to each other, except for CNN, and are between ±0.12 and ±1.85. The results show that the proposed model can detect different attacks with high DR. It can also be seen that the detection in #5 is low. A possible reason for the lower performance of #5 could include differences in training data of the intrinsic property of zero overall theft.

The evaluation of our CNN-based model involves training on a hybrid dataset encompassing various attack types and subsequently testing it on different data comprising new users not present in the training set. Table 6 illustrates the performance of models within this particular scenario. Notably, the CNN+LR classifier demonstrated the highest accuracy, DR, and F1-score. This classifier achieved 95.34% accuracy and 95.01% DR while maintaining a low FPR of only 4.32%. Promising results indicate the effectiveness of our model in detecting thefts from new users by relying on historical data. It shows that the general performance of the model is promising according to the metrics.

Figure 7 presents the ROC graphs presenting TPR against FPR for CNN-based models. The graphs demonstrate the relationship between the ability of models to detect positives and their corresponding false alarm rates. The CNN+LR model achieves an average value of 95.68%, marginally surpassing CNN+RF, which records an average AUC of 95.53%. These AUC values signify the adeptness of both models in correctly classifying instances involving malicious and honest consumers. CNN allows for the extraction of discriminative features from raw data, endowing these models with enhanced generalization capabilities. Furthermore, the CNN-RF model exhibits notable performance, nearing the top performance among the models evaluated. The competitive performance of the CNN+RF model positions it favorably among the hybrid models evaluated for classification efficiency with CNN+LR.

Through the application of appropriate classifying techniques, our model is robust against FDI attacks and non-malicious changes in consumption patterns and therefore acquires high DR and low false alarm (FA).

### 4.4. Comparision with Different Models

Some studies have compared their proposed model with those running on different datasets. It is clear that this is not a healthy comparison. Therefore, we preferred studies using the ISSDA dataset in our comparisons. Also, in some studies, the success of the model was evaluated based on a single attack vector. Since this does not provide a generalizable model, we evaluate our model according to *hybrid dataset* performance metrics.

We make comparisons based on the most commonly used performance metrics in the literature, which are shown in Table 7 with their percentages. Our CNN-based hybrid model achieved the highest performance in accuracy, AUC, F1-score, and DR metrics. Since our FPR metric is relatively high, our HD value is slightly lower than MP-ANN [27]. When (-) is used, it means there is no information about that metric in the related article.

## 5. Conclusions

In this paper, robust CNN-based models are investigated to detect energy theft using real power consumption data of 2104 residential consumers. The models rely on predicting both honest and malicious energy usage pattern of users. The CNN is used to identify patterns of attacks within consumer behavior. In particular, six attack vectors are employed to generate the malicious readings from honest samples from a real energy consumption dataset. Then, CNN-based detectors are proposed for detecting energy theft. Furthermore, the GAN algorithm is adopted to handle the imbalanced data issue. In the CNN-based model, ML algorithms such as SVM, RF, LR, KNN, and DT are applied to the problem as a benchmark. The results show that the proposed CNN+LR and CNN+RF models are considerably promising classification methods in energy theft detection. The hybrid model can automatically extract features with CNN and combine the advantages of RF or LR into the model. In addition, the results indicate that the specified attack vector detectors have better performance compared to the energy-theft-detection models for all attack vectors. The results show the efficiency of models in accurately identifying various attack vectors, marking it as a promising solution for addressing energy theft issues in real-world applications, particularly within the domain of consumer-related threats. As part of future work, investigating the extension of the proposed CNN-based hybrid models to high-consumption data from industrial users is worth investigating.

## Figures and Tables

**Figure 1 sensors-24-01148-f001:**
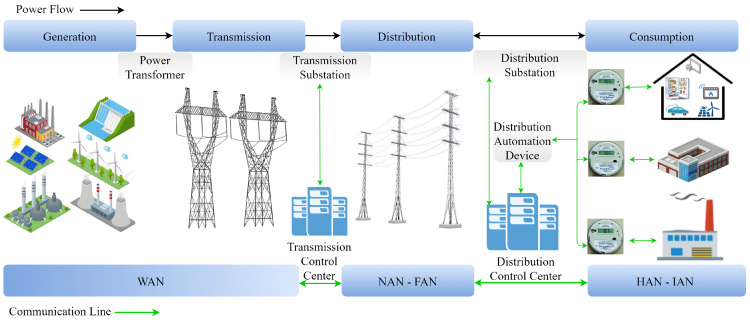
Overall structure of the smart grid environment [20].

**Figure 2 sensors-24-01148-f002:**
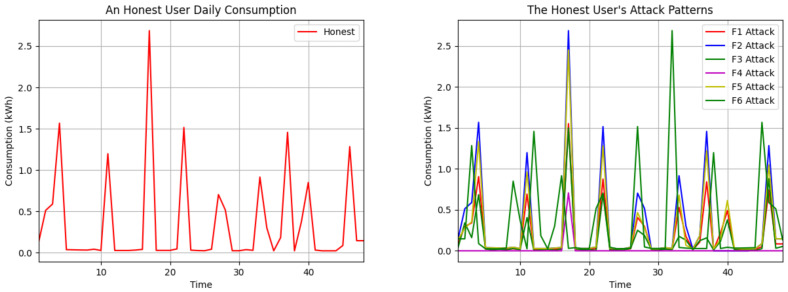
Visualization of mentioned attack vectors in the daily consumption data of an honest user.

**Figure 3 sensors-24-01148-f003:**
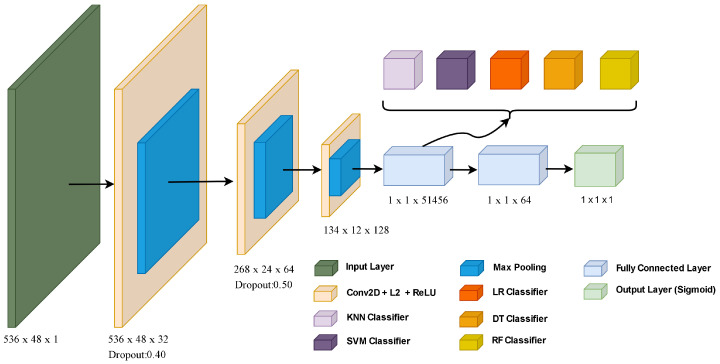
The architecture of the proposed CNN-based model.

**Figure 4 sensors-24-01148-f004:**
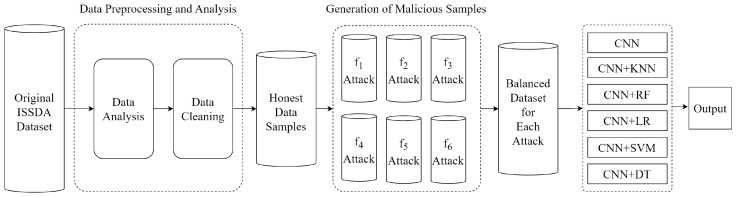
Flowchart of energy theft detection for each attack vector.

**Figure 5 sensors-24-01148-f005:**
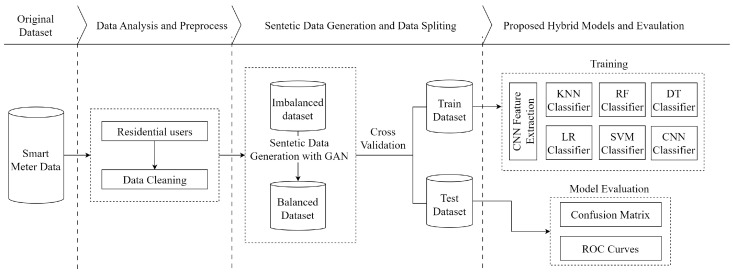
Flowchart of energy theft detection for all attack vectors.

**Figure 6 sensors-24-01148-f006:**
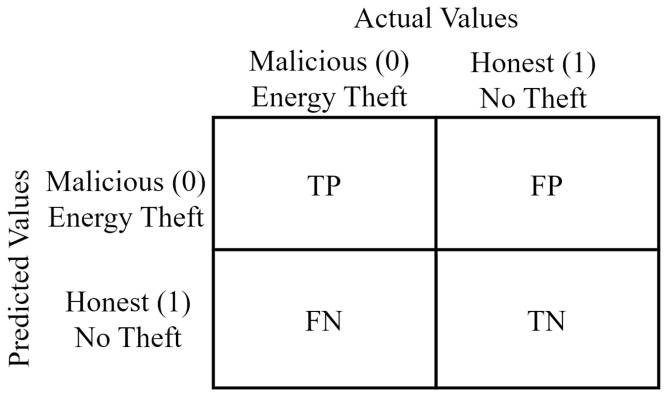
Confusion matrix utilized in energy theft detection.

**Figure 7 sensors-24-01148-f007:**
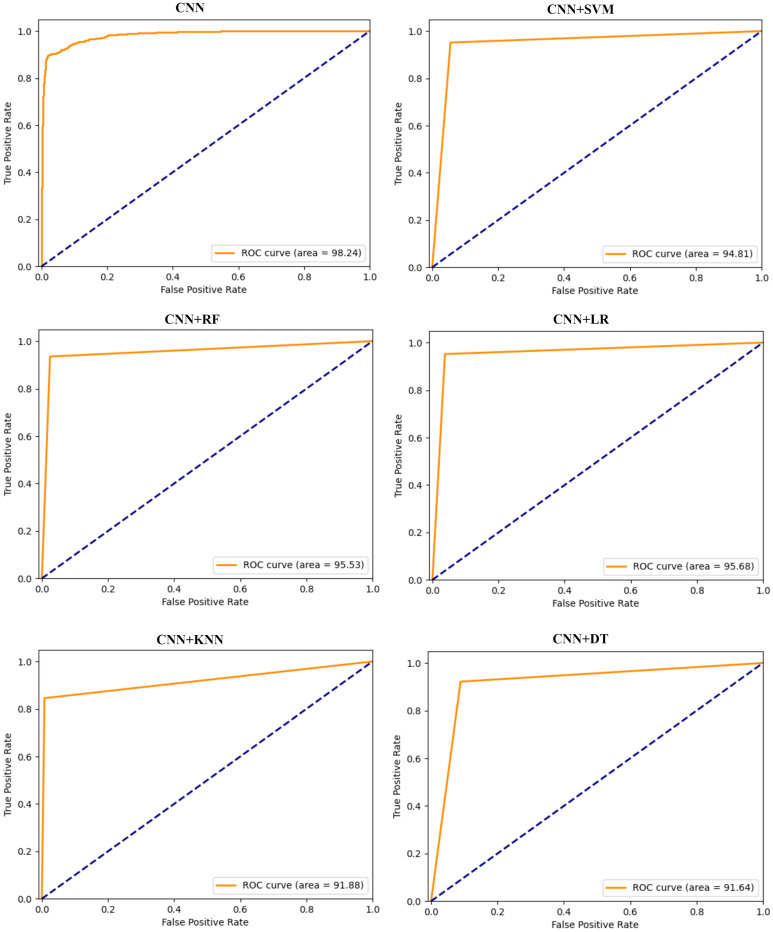
ROC graphs of CNN-based hybrid models on the *hybrid dataset*.

**Table 1 sensors-24-01148-t001:** Literature overview on energy theft detection based on consumption data.

Ref.	Year	Platform	Proposed Model	Dataset	Accuracy	Presented Main Contribution
[31]	2019	N/A	CNN-LSTM based	SGCC	89	The irregular and abnormal consumption patterns of consumers are analyzed
[34]	2018	N/A	Clustering based	ISSDA	74 (AUC)	Malicious examples are not needed to train the method for future detection
[28]	2020	Python 3.x	CNN-GRU-PSO	SGCC	89	Preprocessing steps, feature selection, feature extraction, and classification are performed using a lot of techniques and the proposed model outperforms imbalancing issue
[27]	2020	N/A	MP-ANN	ISSDA	93.4 (DR)	Self-organizing is used for clustering the consumers according to similar consumption patterns, i.e., classification as honest or malicious. The number of transformers that have suspect consumers is reduced without the need to install measurement units on all transformers
[32]	2020	Python 3.x	CNN-LSTM	SGCC	87.9	An efficient solution to overcome imbalanced data, overfitting, and high-dimensional data limitations is introduced
[24]	2016	N/A	DT-SVM	OpenEnergy	92.5	The newly proposed system exhibits the capability to accurately identify instances of energy theft in real time across all stages of power transmission and distribution
[29]	2018	Python 3.x	GRU (RNN-based)	ISSDA	92.5 (DR)	Temporal patterns are utilized in energy consumption, and a GRU-based RNN enhances detection performance, optimizing hyperparameters through a random search analysis in the learning phase
[23]	2016	N/A	SVM-based	ISSDA	94 (DR)	Six different attack vectors are designed to obtain manipulated consumption data
[25]	2021	Python 3.x	Ensemble ML	ISSDA	90 (AUC)	Data pre-processing is used to address imbalanced data with SMOTE and Near-miss techniques, achieving optimal detection rates through bagging-type ensemble ML demonstrated with diverse consumer samples
[30]	2018	N/A	Wide and Deep CNN	SGCC	80 (AUC)	Unlike existing methods tailored for one-dimensional data, wide and deep CNN handles detecting electricity theft by effectively capturing both periodic and non-periodic consumption patterns in two-dimensional data
[33]	2020	N/A	LSTM-based	SGCC	93.6	A new technique is devised to streamline data, enhancing usability and facilitating the extraction of meaningful insights from the dataset
[26]	2021	Python 3.x	Neural Network	Grid LabD Tool	93	A novel method is introduced for detecting electricity theft, focusing on “balance attacks” with prosumers manipulating readings for total aggregated balance. A cluster-based detection model is introduced as a middle-ground approach, bridging the gap between using a single model for all users and individual models for each user
[36]	2021	Matlab2019	CNN-WeightedRF	Mathpower Tool	95.71	An FDI intrusion-detection model combining CNN and weighted RF is able to detect the spurious data more accurately compared with other detection models
[37]	2015	N/A	SVM	ISSDA	75.8	The classification models simplify a demand-side management study, analyze tariff methods, and offer insights for policymakers
[38]	2010	VisualBasic	SVM	Tenaga Nasional	60	This work aims to aid Tenaga Nasional Berhad Distribution in Malaysia to reduce NTLs within the distribution sector caused by electricity theft
[39]	2018	Python 3.x	DNN-based	ISSDA	92.6 (DR)	This work proposes a DNN-based customer-specific detector that can mitigate electricity theft cyber-attacks
[40]	2017	N/A	Density-based clustering	ISSDA	93.2	This work exhibits superior performance compared to alternative methods across nearly all categories of theft
[41]	2022	Python 3.x	Attention LSTM Inception	SGCC	95	This work addresses the elevated FPR issue arising from widespread misclassification, leading to financial burdens
[42]	2022	Python 3.x	KTBoost Classifier	SGCC	93.38	Taking into account all minority sample regions in the dataset, the robust-SMOTE technique generates minority class samples with reduced susceptibility to overfitting and the generation of noisy samples
[43]	2023	Python 3.7	Deep-CNN	Researcher-generated	95	The proposed theft detection method, utilizing the SMOTE technique to generate minority class samples with reduced susceptibility to overfitting and noise, attains the highest accuracy compared to all other studied methods
Our work	2024	Python 3.10	CNN-based	ISSDA	95.34	CNN-based architecture is combined with traditional ML methods. A detector that provides high success in detecting all attack vectors has been designed. The imbalanced data problem was solved using GAN.

**Table 2 sensors-24-01148-t002:** Mathematical representation of cyber attack vectors.

$f_{1}\left(x_{t}\right)=\alpha \times x_{t},$ α= *random*$\left(0.1-0.8\right)$
$f_{2}\left(x_{t}\right)=\left\{\begin{array}{c} 0,\;\left(38\leq x_{t}\leq 43\right)\\ x_{t},\;\;\;otherwise \end{array}\right.$
$f_{3}\left(x_{t}\right)=\lambda_{t}\times x_{t},\;\;\lambda_{t}=$ *random*$\left(0.1-0.8\right)$
$f_{4}\left(x_{t}\right)=\left\{\begin{array}{c} 0,\;\left(x_{t}-\lambda_{t}\right)<0\\ \left(x_{t}-\lambda_{t}\right),\;\;otherwise \end{array}\right.$ $\lambda_{t}=random\left(0.1,\max\left(X\right)\right)$
$f_{5}\left(x_{t}\right)=\left\{\begin{array}{c} x_{t}-mean\left(X\right),\;\left(x_{t}-mean\left(X\right)\right)\geq 0\\ x_{t},\;\;\;otherwise \end{array}\right.$
$f_{6}\left(x_{t}\right)=x_{\left(49-t\right)},\;t=\left\{1,2,3,\ldots ,48\right\}$

**Table 3 sensors-24-01148-t003:** Generated datasets with *f* attacks.

Generated Dataset	$\begin{array}{c} \mathrm{Number}\;\mathrm{of}\;\mathrm{Samples}\\ (\mathrm{Honest}\;+\;\mathrm{Malicious}) \end{array}$	Number of Readings
Honest + $f_{1}$ attack	2104+2104	536 × 48×2104×2
Honest + $f_{2}$ attack	2104+2104	536×48×2104×2
Honest + $f_{3}$ attack	2104+2104	536×48×2104×2
Honest + $f_{4}$ attack	2104+2104	536×48×2104×2
Honest + $f_{5}$ attack	2104+2104	536×48×2104×2
Honest + $f_{6}$ attack	2104+2104	536×48×2104×2
*Hybrid Dataset*	2104×6+2104×6	536×48×2104×12

**Table 4 sensors-24-01148-t004:** Performance metrics obtained from the confusion matrix.

Metric	Equation	Short Description
TPR (DR)	$\frac{\mathrm{TP}}{\mathrm{TP}+\mathrm{FN}}$	The performance of the CNN model in detecting malicious samples.
Specificity	$\frac{\mathrm{TN}}{\mathrm{TN}+\mathrm{FP}}$	The performance of the CNN model in detecting honest samples.
Precision	$\frac{\mathrm{TP}}{\mathrm{TP}+\mathrm{FP}}$	The ratio of malicious samples predicted as malicious to all malicious samples.
F1-score	$\frac{2\mathrm{TP}}{2\mathrm{TP}+\mathrm{FP}+\mathrm{FN}}$	Expressed as the harmonic mean between precision and sensitivity.
Accuracy	$\frac{\mathrm{TP}+\mathrm{TN}}{\mathrm{TP}+\mathrm{FP}+\mathrm{FN}+\mathrm{TN}}$	Measures the overall correctness of predictions by the model.
FPR	$\frac{\mathrm{TP}}{\mathrm{FP}+\mathrm{TN}}$	Measures the proportion of actual honest that were incorrectly classified as malicious samples.

**Table 5 sensors-24-01148-t005:** All evaluation results of the CNN-based models for each *f* attack.

Dataset	Model	Accuracy	Precision	Sensitivity	Specificity	F1-Score	FPR
Honest + $f_{1}$ Attack (#1)	CNN	92.4 ± 3.04	99.71 ± 0.18	87.24 ± 4.72	99.67 ± 0.2	92.99 ± 2.64	0.33 ± 0.2
	CNN+SVM	96.6 ± 0.95	97.2 ± 0.7	96.06 ± 1.22	97.16 ± 0.72	96.62 ± 0.93	2.84 ± 0.72
	CNN+LR	97.15 ± 0.79	98.24 ± 0.7	96.14 ± 0.9	98.2 ± 0.72	97.18 ± 0.77	1.8 ± 0.72
	CNN+RF	97.08 ± 0.79	98.67 ± 0.68	95.63 ± 0.87	98.62 ± 0.71	97.12 ± 0.77	1.38 ± 0.71
	CNN+KNN	94.82 ± 0.78	99.71 ± 0.18	90.84 ± 1.24	99.68 ± 0.2	95.07 ± 0.71	0.32 ± 0.2
	CNN+DT	93.23 ± 0.63	92.92 ± 1.25	93.5 ± 0.78	92.98 ± 1.15	93.2 ± 0.66	7.02 ± 1.15
Honest + $f_{2}$ Attack (#2)	CNN	99.55 ± 0.12	99.81 ± 0.28	99.29 ± 0.45	99.81 ± 0.28	99.55 ± 0.12	0.19 ± 0.28
	CNN+SVM	99.74 ± 0.14	99.76 ± 0.26	99.72 ± 0.28	99.76 ± 0.26	99.74 ± 0.14	0.24 ± 0.26
	CNN+LR	99.67 ± 0.14	99.76 ± 0.37	99.58 ± 0.38	99.76 ± 0.36	99.67 ± 0.14	0.24 ± 0.36
	CNN+RF	99.83 ± 0.14	99.86 ± 0.29	99.81 ± 0.23	99.86 ± 0.28	99.83 ± 0.14	0.14 ± 0.28
	CNN+KNN	98.43 ± 0.43	98.86 ± 0.63	98.03 ± 0.81	98.85 ± 0.63	98.44 ± 0.43	1.15 ± 0.63
	CNN+DT	99.67 ± 0.22	99.81 ± 0.23	99.53 ± 0.42	99.81 ± 0.23	99.67 ± 0.22	0.19 ± 0.23
Honest + $f_{3}$ Attack (#3)	CNN	94.2 ± 2.17	99.86 ± 0.19	89.83 ± 3.29	99.83 ± 0.23	94.55 ± 1.91	0.17 ± 0.23
	CNN+SVM	98.24 ± 0.57	99.1 ± 0.61	97.43 ± 0.55	99.08 ± 0.62	98.26 ± 0.56	0.92 ± 0.62
	CNN+LR	97.96 ± 0.87	99.1 ± 0.55	96.89 ± 1.15	99.07 ± 0.57	97.98 ± 0.85	0.93 ± 0.57
	CNN+RF	98.74 ± 0.24	99.48 ± 0.59	98.04 ± 0.42	99.47 ± 0.59	98.75 ± 0.25	0.53 ± 0.59
	CNN+KNN	94.13 ± 1.85	99.9 ± 0.19	89.66 ± 2.94	99.9 ± 0.21	94.48 ± 1.63	0.1 ± 0.21
	CNN+DT	96.48 ± 0.29	96.25 ± 0.57	96.71 ± 0.44	96.27 ± 0.54	96.47 ± 0.29	3.73 ± 0.54
Honest + $f_{4}$ Attack (#4)	CNN	98.69 ± 0.92	97.91 ± 2.06	99.47 ± 0.32	97.99 ± 1.95	98.67 ± 0.95	2.01 ± 1.95
	CNN+SVM	99.12 ± 0.29	99.24 ± 0.61	99.01 ± 0.37	99.24 ± 0.6	99.12 ± 0.29	0.76 ± 0.6
	CNN+LR	99.31 ± 0.14	99.43 ± 0.41	99.2 ± 0.32	99.43 ± 0.41	99.31 ± 0.14	0.57 ± 0.41
	CNN+RF	99.6 ± 0.18	99.71 ± 0.18	99.48 ± 0.27	99.71 ± 0.18	99.6 ± 0.18	0.29 ± 0.18
	CNN+KNN	98.57 ± 0.67	97.72 ± 1.06	99.42 ± 0.36	97.77 ± 1.01	98.56 ± 0.69	2.23 ± 1.01
	CNN+DT	99.05 ± 0.21	99.14 ± 0.12	98.96 ± 0.32	99.14 ± 0.12	99.05 ± 0.21	0.86 ± 0.12
Honest +$f_{5}$ Attack (#5)	CNN	82.46 ± 4.1	82.09 ± 13.91	83.46 ± 3.95	83.98 ± 8.75	81.76 ± 6.71	16.02 ± 8.75
	CNN+SVM	82.08 ± 0.32	80.94 ± 1.47	82.86 ± 1.09	81.39 ± 0.92	81.87 ± 0.39	18.61 ± 0.92
	CNN+LR	83.2 ± 0.71	83.36 ± 0.99	83.14 ± 1.73	83.32 ± 0.51	83.23 ± 0.48	16.68 ± 0.51
	CNN+RF	83.32 ± 0.92	86.41 ± 0.99	81.42 ± 1.62	85.52 ± 0.74	83.82 ± 0.72	14.48 ± 0.74
	CNN+KNN	76.14 ± 0.35	75.71 ± 2.58	76.39 ± 1.63	76.05 ± 1.23	75.99 ± 0.61	23.95 ± 1.23
	CNN+DT	75.9 ± 1.78	75.52 ± 2.52	76.12 ± 1.88	75.74 ± 2.05	75.8 ± 1.86	24.26 ± 2.05
Honest + $f_{6}$ Attack (#6)	CNN	98.27 ± 0.46	98.15 ± 0.66	98.38 ± 0.65	98.16 ± 0.65	98.26 ± 0.46	1.84 ± 0.65
	CNN+SVM	97.24 ± 0.68	97.24 ± 0.95	97.25 ± 0.97	97.25 ± 0.92	97.24 ± 0.68	2.75 ± 0.92
	CNN+LR	97.98 ± 0.56	97.96 ± 0.56	98.01 ± 0.98	97.96 ± 0.54	97.98 ± 0.55	2.04 ± 0.54
	CNN+RF	98.46 ± 0.49	98.38 ± 0.79	98.53 ± 0.74	98.39 ± 0.77	98.45 ± 0.49	1.61 ± 0.77
	CNN+KNN	97.01 ± 0.56	97.01 ± 1.07	97.01 ± 0.75	97.02 ± 1.02	97 ± 0.56	2.98 ± 1.02
	CNN+DT	95.75 ± 0.97	95.77 ± 0.55	95.74 ± 1.49	95.77 ± 0.57	95.75 ± 0.94	4.23 ± 0.57

**Table 6 sensors-24-01148-t006:** Performance metrics of the proposed CNN-based models for the *hybrid dataset*.

Dataset	Model	Accuracy	Precision	Sensitivity	Specificity	F1-Score	FPR
	CNN	92.26 ± 0.94	89.54 ± 1.69	94.69 ± 0.74	90.11 ± 1.44	92.04 ± 1.02	9.89 ± 1.44
	CNN+SVM	94.24 ± 0.41	94.23 ± 0.42	94.25 ± 0.59	94.24 ± 0.4	94.24 ± 0.4	5.76 ± 0.4
*Hybrid*	CNN+LR	95.34 ±0.25	95.71 ± 0.34	95.01 ± 0.3	95.68 ± 0.33	95.36 ± 0.25	4.32 ± 0.33
*Dataset*	CNN+RF	94.93 ± 0.38	97.34 ± 0.43	92.87 ± 0.65	97.21 ± 0.44	95.05 ± 0.36	2.79 ± 0.44
	CNN+KNN	90.43 ± 0.96	98.24 ± 1.05	84.99 ± 1.37	97.94 ± 1.16	91.13 ± 0.84	2.06 ± 1.16
	CNN+DT	91.11 ± 0.42	90.56 ± 0.76	91.59 ± 0.86	90.67 ± 0.64	91.06 ± 0.41	9.33 ± 0.64

**Table 7 sensors-24-01148-t007:** The performance of our model and others in the literature.

Ref.	Dataset	Proposed Model	Acc	AUC	F1	DR (TPR)	FA (FPR)	HD (TPR-FPR)
[34]	ISSDA	Clustering-based	-	-	-	63.6	24.3	-
[27]	ISSDA	MP-ANN	-	-	-	93.4	1.9	91.5
[29]	ISSDA	GRU RNN-based	-	-	-	92.5	5	87.5
[23]	ISSDA	SVM-based	-	-	-	94	11	83
[25]	ISSDA	Ensemble ML	-	90	-	-	-	-
[37]	ISSDA	SVM	75.8	80.2	-	-	-	-
[39]	ISSDA	DNN-based	-	-	-	92.6	2.3	90.3
[40]	ISSDA	Density-based clustering	93.2	74.3	32.2	-	-	-
[46]	ISSDA	Semi-supervised	-	84.2	73.3	-	-	-
[51]	ISSDA	Feedforward ANN based	93.36	-	-	92.56	5.84	86.72
Our work	ISSDA	CNN-based hybrid	95.34	95.68	95.36	95.01	4.32	90.69

## Data Availability

Previously reported ISSDA dataset was used to support this study and available at https://www.ucd.ie/issda/data/ (accessed on 1 January 2024). The dataset is cited at relevant place in the text.

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
