# Peer review of "Smart Grid Security: An Effective Hybrid CNN-Based Approach for Detecting Energy Theft Using Consumption Patterns"

_sensors, 2024, doi:10.3390/s24041148_

Round 1
Reviewer 1 Report
Comments and Suggestions for Authors
please see attached file

please see attached file
Author Response
Authors’ Response to the Review Comments
Journal : MDPI, Sensors
Manuscript ID : Sensors-2822818
Title of Paper : Detecting Energy Theft in Smart Grids: An Effective Hybrid CNN-based Approach
Using Consumption Patterns
Authors : M.Zekeriya Gunduz, Resul Das
Date Sent : January 24, 2024
First of all, We would like to very much thankful to the Editor and Reviewers for their deep and thorough review. We have revised our existing research article in light of their useful suggestions and comments. We hope that this major revision has improved the paper to a level of your satisfaction.
Number wise answers to their specific comments /suggestions /queries are as follows.
Responses to Comments of Reviewer 1
Reviewer 1’ comments: A Hybrid CNN-based approach That Effectively Detects Energy Theft in Smart Grids by Analyzing Consumption Patterns is presented in this manuscript. The authors ought to respond to the following remarks, notwithstanding the fact that the scientific contribution of the manuscript is adequately defined but not explicit for the readership:
Response: First of all, we would like to thank you very much for your valuable view and your comment that adds value to the article. As evaluated by the reviewer, we wanted to make a well-organized paper of our study about energy theft detection in the smart grid. So we are grateful for this evaluation.
Reviewer 1’ comments: Could you rewrite the contribution part? It is not clear enough. Readers require further clarification.
Response: First of all, I appreciate the reviewer’s insightful comment. Thanks for the valuable comments.
The contribution part has been rewritten under the title "1.1. Research Contributions" in bullet points to make it clearer and more understandable.
Reviewer 1’ comments: What do the acronyms that follow denote? NAN-FAN-HAN-BAN-IAN, NTL, SMOTE, SVM, RF, DT. Please add them to the manuscript. (Not only in abstract)
Response: Thanks for the valuable comments. We greatly appreciate the reviewer’s efforts to carefully review the paper and the valuable suggestions offered. All acronyms used in the text have been reviewed and explained where they are first used. Also, the abbreviations section has been added into the page 17.
Reviewer 1’ comments: The related works section seems like a survey, not a review of previous works. Please reorganize, critically review them again, and show their deficits compared to your current work.
Response: Thanks for the valuable comment. We have revised the section to provide a more in-depth analysis of each previous work. We have taken a closer look at their methodologies, findings, and contributions, and critically assessed their limitations. In doing so, we have explicitly highlighted the deficits in these works when compared to our current research.
Reviewer 1’ comments: According to my point of view, Table 1 has the potential to be enhanced, and additional criteria, metrics, or characteristics might be included. Table 1 seems to include only survey data, but there is no critical review of previous works.
Response: Thanks for the valuable comment. In Table 1, the presented main contributions emphasized by relevant researchers are presented under the title 'Presented Main Contribution.' Additionally, our study has been critically examined in comparison to the most significant of these works, and a detailed comparison has been provided in Table 1 and Table 7 based on various metrics.
Reviewer 1’ comments: Could you please define what the hybrid models are in the context of this research paper as well as for deep learning? What do you mean by “CNN-based hybrid”? Do you mean that you are utilizing a hybrid dataset or a hybrid methodology? Please clarify, explain, and detail it.
Response: Thanks for the valuable comment. We used CNN-shallow machine learning hybrid structures on electricity consumption datasets to develop energy theft detector. We combined the CNN with shallow machine learning techniques on the electricity consumption dataset to solve the classification problem. This encouraged us to exploit this hybrid structure to detect electricity theft by analyzing the irregular and abnormal consumption patterns of consumers. CNN+(SVM/RF/DT/KNN/LR) are the hybrid structures. CNN is for feature extraction and the others for classification. We mean utilizing a hybrid methodology. It is explained in detail in the introduction section rows 78-84.
Reviewer 1’ comments: The Author claims that α and λ are randomly generated numbers between 0.1 and 0.8. Why did you use this range? Are there any references for validation using this number range?
Response: Thanks for the valuable comment. Suggested attack vectors by Jokar et al. in 2016 are notably implemented in the literature with the specified α and λ values. The stated explanations are expressed in the text page 8 rows 280-287.
- Jokar, N. Arianpoo and V. C. M. Leung, "Electricity Theft Detection in AMI Using Customers’ Consumption Patterns," in IEEE Transactions on Smart Grid, vol. 7, no. 1, pp. 216-226, Jan. 2016,
doi: 10.1109/TSG.2015.2425222.
Reviewer 1’ comments: What are assumptions of attack vectors? How did you decide these attack vectors? Why? Any references? Please explain it.
Response: Thanks for the valuable comment. The attack vectors proposed by Jokar et al. since 2016 have been notably implemented in the literature. Additionally, many researchers have generated variations of these attack vectors. Some references added accordingly. The stated explanations are expressed in the text page 7, rows 280-281.
Viegas, Joaquim L., Paulo R. Esteves, and Susana M. Vieira. "Clustering-based novelty detection for identification of non-technical losses." International Journal of Electrical Power & Energy Systems 101 (2018): 301-310.
https://doi.org/10.1016/j.ijepes.2018.03.031
de Souza, Matheus Alberto, et al. "Detection and identification of energy theft in advanced metering infrastructures." Electric Power Systems Research 182 (2020): 106258. https://doi.org/10.1016/j.epsr.2020.106258
- Nabil, M. Ismail, M. Mahmoud, M. Shahin, K. Qaraqe and E. Serpedin, "Deep Recurrent Electricity Theft Detection in AMI Networks with Random Tuning of Hyper-parameters," 2018 24th International Conference on Pattern Recognition (ICPR), Beijing, China, 2018, pp. 740-745,
doi: 10.1109/ICPR.2018.8545748.
Gunturi, Sravan Kumar, and Dipu Sarkar. "Ensemble machine learning models for the detection of energy theft." Electric Power Systems Research 192 (2021): 106904.
https://doi.org/10.1016/j.epsr.2020.106904
- Alromih, J. A. Clark and P. Gope, "Electricity Theft Detection in the Presence of Prosumers Using a Cluster-based Multi-feature Detection Model," 2021 IEEE International Conference on Communications, Control, and Computing Technologies for Smart Grids (SmartGridComm), Aachen, Germany, 2021, pp. 339-345,
doi: 10.1109/SmartGridComm51999.2021.9632322.
- Ismail, M. Shahin, M. F. Shaaban, E. Serpedin and K. Qaraqe, "Efficient detection of electricity theft cyber attacks in AMI networks," 2018 IEEE Wireless Communications and Networking Conference (WCNC), Barcelona, Spain, 2018, pp. 1-6,
doi: 10.1109/WCNC.2018.8377010.
Reviewer 1’ comments: Could you kindly review the language used in some sections of the manuscript, as well as punctuation and typos?
Response: The entire study has been corrected by a native English speaker from the beginning to the end. The article was rearranged considering English grammar, spelling and sentence structure.
Finally and most importantly, the paper was checked again to improve its grammar and as well as removing the typographical errors in the revised version. In addition, all references cited in the article were revised, the spelling deficiencies were fixed, and it was automatically rewritten from the .bib file by using Zotero. Moreover, the article was completely revised and written in LaTeX format.
Thank you so much for your contributions.
Authors.
Reviewer 2 Report
Comments and Suggestions for Authors
The authors have proposed CNN based hybrid Intrusion Detection Systems to detect data tampering using six attack vectors causing FDI. The work is interesting; however, the manuscript is needed to be revised as per comments extended below:
1. The same authors have presented a DNN based model in the following published work.
“M. Z. GÜNDÜZ and R. DAÅž, “An effective DNN-based Approach for Detecting Energy Theft in Smart Grids through User Consumption Patterns”, TJNS, vol. 12, no. 4, pp. 163–170, 2023, doi: 10.46810/tdfd.1383065.”
The authors claimed an overall accuracy of 97.4% in their already published work, in the current work the overall accuracy is 95.34%. How is it justified as compared to their already published work?
2. Page 6, line 239: Authors have stated that:
“There are some general attack vectors that used in literature. We used six different attack vectors on honest consumption data”
No reference and justification of choosing these attack vectors have been provided. In addition, these sentences need grammatical correction.
3. Page 2, lines 65-70: The section numbering and paper structure is totally mismatching with actual formatting of the manuscript.
4. The section ‘0’ introduction is insufficient; it is better to merge the section ‘0’ and section 1 and make the overall flow of the work proper.
5. Page 5, line 159: Authors have claimed that their work is original research article and at the same time it has the characteristics of a review paper. This is very confusing as the work lacks the components of a review paper and this claim should be removed in order to be specific about your work.
6. Page 6, line 233: The sentence “The created matrixes are given as input to the CNN network” should be corrected as the word “matrixes” is not commonly used.
7. Page 7, line 256: The sentence “The impact of these six attack vectors on an daily use of honest customer can be seen in Figure 2.” Needs correction.
8. Page 9, line 310: The sentence “where δ and ∗ show the activation function and convolution, respectively. and Show learnable parameters.” Needs correction.
9. The conclusion section is just resembling the abstract, it is needed to be rewritten in a more professional style.
Comments on the Quality of English Languageplease see the authors comments
Author Response
Authors’ Response to the Review Comments
Journal : MDPI, Sensors
Manuscript ID : Sensors-2822818
Title of Paper : Detecting Energy Theft in Smart Grids: An Effective Hybrid CNN-based Approach
Using Consumption Patterns
Authors : M.Zekeriya Gunduz, Resul Das
Date Sent : January 24, 2024
First of all, We would like to very much thankful to the Editor and Reviewers for their deep and thorough review. We have revised our existing research article in light of their useful suggestions and comments. We hope that this major revision has improved the paper to a level of your satisfaction.
Number wise answers to their specific comments /suggestions /queries are as follows.
Responses to Comments of Reviewer 2
Reviewer 2’ comments: The authors have proposed CNN based hybrid Intrusion Detection Systems to detect data tampering using six attack vectors causing FDI. The work is interesting; however, the manuscript is needed to be revised as per comments extended below:
Response: First of all, we would like to thank you very much for your valuable view and your comment that adds value to the article. This precious review and valuable comments shed light on our work. Thank you so much again.
Reviewer 2’ comments: The same authors have presented a DNN based model in the following published work.
“M. Z. GÜNDÜZ and R. DAÅž, “An effective DNN-based Approach for Detecting Energy Theft in Smart Grids through User Consumption Patterns”, TJNS, vol. 12, no. 4, pp. 163–170, 2023, doi: 10.46810/tdfd.1383065.”
The authors claimed an overall accuracy of 97.4% in their already published work, in the current work the overall accuracy is 95.34%. How is it justified as compared to their already published work?
Response: We appreciate the reviewer’s insightful comment. Thanks for the valuable comment.
While only 2 attack vectors were used in our previously published study, 6 attack vectors were used in our current study. While only Neural Network was used in the previous study, CNN was used in the current study. While the previous study highlighted the detection of a single attack vector, in the current study a system that detects all 6 different attack vectors has been developed. Finally, the attack vector, which was detected with a rate of 97.4% in the previous study, was detected with a success rate of over 99% in our current study.
Reviewer 2’ comments: Page 6, line 239: Authors have stated that:
“There are some general attack vectors that used in literature. We used six different attack vectors on honest consumption data.” No reference and justification of choosing these attack vectors have been provided. In addition, these sentences need grammatical correction.
Response: Thanks for the valuable comment. The attack vectors proposed by Jokar et al. since 2016 have been notably implemented in the literature. Additionally, many researchers have generated variations of these attack vectors. Some references added accordingly. The stated explanations are expressed in the text. Also, grammatical problems were corrected.
Viegas, Joaquim L., Paulo R. Esteves, and Susana M. Vieira. "Clustering-based novelty detection for identification of non-technical losses." International Journal of Electrical Power & Energy Systems 101 (2018): 301-310. https://doi.org/10.1016/j.ijepes.2018.03.031
de Souza, Matheus Alberto, et al. "Detection and identification of energy theft in advanced metering infrastructures." Electric Power Systems Research 182 (2020): 106258. https://doi.org/10.1016/j.epsr.2020.106258
- Nabil, M. Ismail, M. Mahmoud, M. Shahin, K. Qaraqe and E. Serpedin, "Deep Recurrent Electricity Theft Detection in AMI Networks with Random Tuning of Hyper-parameters," 2018 24th International Conference on Pattern Recognition (ICPR), Beijing, China, 2018, pp. 740-745, doi: 10.1109/ICPR.2018.8545748.
Gunturi, Sravan Kumar, and Dipu Sarkar. "Ensemble machine learning models for the detection of energy theft." Electric Power Systems Research 192 (2021): 106904. https://doi.org/10.1016/j.epsr.2020.106904
- Alromih, J. A. Clark and P. Gope, "Electricity Theft Detection in the Presence of Prosumers Using a Cluster-based Multi-feature Detection Model," 2021 IEEE International Conference on Communications, Control, and Computing Technologies for Smart Grids (SmartGridComm), Aachen, Germany, 2021, pp. 339-345, doi: 10.1109/SmartGridComm51999.2021.9632322.
- Ismail, M. Shahin, M. F. Shaaban, E. Serpedin and K. Qaraqe, "Efficient detection of electricity theft cyber attacks in AMI networks," 2018 IEEE Wireless Communications and Networking Conference (WCNC), Barcelona, Spain, 2018, pp. 1-6, doi: 10.1109/WCNC.2018.8377010.
Reviewer 2’ comments: Page 2, lines 65-70: The section numbering and paper structure is totally mismatching with actual formatting of the manuscript.
Response: Thanks for the valuable comment. The section numbering and paper structure was totally corrected according to the actual formatting of the manuscript.
Reviewer 2’ comments: The section ‘0’ introduction is insufficient; it is better to merge the section ‘0’ and section 1 and make the overall flow of the work proper.
Response: We greatly appreciate the reviewer’s efforts to carefully review the paper and the valuable suggestions offered. Thank you so much again. The introduction section has been revised and expanded. Additionally, the first and second sections were restructured.
Reviewer 2’ comments: Page 5, line 159: Authors have claimed that their work is original research article and at the same time it has the characteristics of a review paper. This is very confusing as the work lacks the components of a review paper and this claim should be removed in order to be specific about your work.
Response: Thank you for the comment. The relevant claim was removed.
Reviewer 2’ comments: Page 6, line 233: The sentence “The created matrixes are given as input to the CNN network” should be corrected as the word “matrixes” is not commonly used.
Response: Thanks for the valuable comment. It was corrected and highlighted.
Reviewer 2’ comments: Page 7, line 256: The sentence “The impact of these six attack vectors on an daily use of honest customer can be seen in Figure 2.” Needs correction.
Response: Thanks for the valuable comment. It has been corrected. Page 8, line 299.
Reviewer 2’ comments: Page 9, line 310: The sentence “where δ and ∗ show the activation function and convolution, respectively. and Show learnable parameters.” Needs correction.
Response: We greatly appreciate the reviewer’s efforts to carefully review the paper and the valuable suggestions offered. Thank you so much again. It has been corrected. Page 10, line 355-356
Reviewer 2’ comments: The conclusion section is just resembling the abstract, it is needed to be rewritten in a more professional style.
Response: Thanks for the valuable comment. The conclusion section has been revised and rewritten in a more professional style.
Finally and most importantly, the paper was checked again to improve its grammar and as well as removing the typographical errors in the revised version. In addition, all references cited in the article were revised, the spelling deficiencies were fixed, and it was automatically rewritten from the .bib file by using Zotero. Moreover, the article was completely revised and written in LaTeX format.
Thank you so much for your contributions.
Authors.
Reviewer 3 Report
Comments and Suggestions for Authors
- 1. The paper represents an interesting and relevant topic. 2. The possibilities of increasing the level of security in smart grids are considered as well as the protection of critical infrastructure is studied. 3. An attractive feature of the study is the application of deep learning (DL) and degenerate adversarial network (GAN) techniques. Although the GANs have been in use since 2014 they are still considered to be a powerful tool providing visual interpretation of complex environments. It is a special class of Artificial Intelligence (AI) algorithms used in unsupervised machine learning, implemented as a system of two neural networks competing with each other. 4. The reviewed study applies the novel machine learning methods in integration with other methods. For example,the method of selecting metrics to evaluate the effectiveness of the author's model for managing potential attacks on smart grid is considered to be relevant to the modern challenges. 5. Overall, the presented work has novelty in the proposed idea of GAN application. All the presented conclusions are valid and correct. 6. The scientific paper can be recommended for publication, but English should be checked and edited for avoiding misprints and misunderstanding.
- All the conclusions are consistent with the arguments presented in the paper. The conclusions are substantiated by available references and figures. However, some figures should be detailed and specified (Fig.7.p.14). The explanations and description should be introduced after the figure, due to the lack of difference in visualization making the semantic interpretation difficult.
Comments on the Quality of English Language
English should be checked and edited for avoiding misprints and misunderstanding.
Author Response
Authors’ Response to the Review Comments
Journal : MDPI, Sensors
Manuscript ID : Sensors-2822818
Title of Paper : Detecting Energy Theft in Smart Grids: An Effective Hybrid CNN-based Approach
Using Consumption Patterns
Authors : M.Zekeriya Gunduz, Resul Das
Date Sent : January 24, 2024
First of all, We would like to very much thankful to the Editor and Reviewers for their deep and thorough review. We have revised our existing research article in light of their useful suggestions and comments. We hope that this major revision has improved the paper to a level of your satisfaction.
Number wise answers to their specific comments /suggestions /queries are as follows.
Responses to Comments of Reviewer 3
Reviewer 3’ comments:
- The paper represents an interesting and relevant topic.
- The possibilities of increasing the level of security in smart grids are considered as well as the protection of critical infrastructure is studied.
- An attractive feature of the study is the application of deep learning (DL) and degenerate adversarial network (GAN) techniques. Although the GANs have been in use since 2014 they are still considered to be a powerful tool providing visual interpretation of complex environments. It is a special class of Artificial Intelligence (AI) algorithms used in unsupervised machine learning, implemented as a system of two neural networks competing with each other.
- The reviewed study applies the novel machine learning methods in integration with other methods. For example, the method of selecting metrics to evaluate the effectiveness of the author's model for managing potential attacks on smart grid is considered to be relevant to the modern challenges.
- Overall, the presented work has novelty in the proposed idea of GAN application. All the presented conclusions are valid and correct.
Response: First of all, we would like to thank you very much for your valuable view and your comment that adds value to the article. This precious review and valuable comments shed light on our work. Thank you so much.
Reviewer 3’ comments:
- The scientific paper can be recommended for publication, but English should be checked and edited for avoiding misprints and misunderstanding.
Response: Thank you for the comment. The article was rearranged considering English grammar, spelling and sentence structure.
Reviewer 3’ comments: All the conclusions are consistent with the arguments presented in the paper. The conclusions are substantiated by available references and figures.
Response: The authors thank the referee for their valuable comments.
Reviewer 3’ comments: Some figures should be detailed and specified (Fig.7.p.14). The explanations and description should be introduced after the figure, due to the lack of difference in visualization making the semantic interpretation difficult.
Response: Thanks for the comment. The figures have been detailed. They are specified within the text. The same visualization features have been used for all figures. Figure explanations have been provided after placement.
Finally and most importantly, the paper was checked again to improve its grammar and as well as removing the typographical errors in the revised version. In addition, all references cited in the article were revised, the spelling deficiencies were fixed, and it was automatically rewritten from the .bib file by using Zotero. Moreover, the article was completely revised and written in LaTeX format.
Thank you so much for your contributions.
Authors.
Reviewer 4 Report
Comments and Suggestions for Authors
The authors proposed a mechanism for detection energy theft in smart grids. The manuscript is almost well-organized, but the following concerns should be addressed.
- Some methodologies for detecting energy theft attacks are outlined in the abstract, yet the discussion also addresses the challenges associated with these approaches.
- Some typos in the manuscript need to be corrected.
- Research gaps are not clear enough.
- It would be preferable to revise the title, "RELATED WORKS AND CONTRIBUTIONS."
- The contribution section requires significant revision.
- How can we verify the statement "We address the challenges posed by zero-day attacks"?
- Is your ISSDA dataset authentic? If so, could you clarify why you make statements like "Obtaining real datasets to detect energy theft is a challenge"?
- Please revise some sentences, like "Figure 4 is explained as follows:".
- The statement "SVM, LR, RF, KNN, DT algorithms were used with CNN for classification" lacks clarity. Are these classifiers essential for CNN classification? Additional details on the rationale behind combining these approaches are necessary.
Comments on the Quality of English LanguageThe authors have introduced a mechanism for detecting energy theft in smart grids. While the manuscript is promising, some revisions are needed.
Author Response
Authors’ Response to the Review Comments
Journal : MDPI, Sensors
Manuscript ID : Sensors-2822818
Title of Paper : Detecting Energy Theft in Smart Grids: An Effective Hybrid CNN-based Approach
Using Consumption Patterns
Authors : M.Zekeriya Gunduz, Resul Das
Date Sent : January 24, 2024
First of all, We would like to very much thankful to the Editor and Reviewers for their deep and thorough review. We have revised our existing research article in light of their useful suggestions and comments. We hope that this major revision has improved the paper to a level of your satisfaction.
Number wise answers to their specific comments /suggestions /queries are as follows.
Responses to Comments of Reviewer 4
Reviewer 4’ comments: The authors proposed a mechanism for detection energy theft in smart grids. The manuscript is almost well-organized, but the following concerns should be addressed.
Response: First of all, we would like to thank you very much for your valuable view and your comment that adds value to the article. As evaluated by the reviewer, we wanted to make a well-organized paper of our study about energy theft detection in the smart grid. So we are grateful for this evaluation.
Reviewer 4’ comments: Some methodologies for detecting energy theft attacks are outlined in the abstract, yet the discussion also addresses the challenges associated with these approaches.
Response: Thanks for the valuable comment. The abstract has been revised again to avoid mentioned complexity.
Reviewer 4’ comments: Some typos in the manuscript need to be corrected.
Response: Thanks for the valuable comments. The manuscript has been reviewed again in terms of grammar and typos.
Reviewer 4’ comments: Research gaps are not clear enough.
Response: Thanks for the valuable comments. Sections 1 and 2 have been revised and structurally modified. Gaps in the literature have been clarified.
Reviewer 4’ comments: It would be preferable to revise the title, "RELATED WORKS AND CONTRIBUTIONS."
Response: Thanks for the valuable comments. The title has been revised.
Reviewer 4’ comments: The contribution section requires significant revision.
Response: Thanks for the valuable comments. The contribution section has been revised and clarified.
Reviewer 4’ comments: How can we verify the statement "We address the challenges posed by zero-day attacks"?
Response: Thanks for the valuable comments. The six attack vectors are presented by Jokar et al. and interestingly used in the literature, aim to reduce data consumption. Various versions of these vectors are also utilized. Therefore, attacks that alter user consumption patterns are considered zero-day attacks. Another goal of the intrusion detection system is to uncover unknown versions of attack vectors by leveraging recognized attack vectors.
- Jokar, N. Arianpoo and V. C. M. Leung, "Electricity Theft Detection in AMI Using Customers’ Consumption Patterns," in IEEE Transactions on Smart Grid, vol. 7, no. 1, pp. 216-226, Jan. 2016,
doi: 10.1109/TSG.2015.2425222.
Reviewer 4’ comments: Is your ISSDA dataset authentic? If so, could you clarify why you make statements like "Obtaining real datasets to detect energy theft is a challenge"?
Response: Thanks for the valuable comments. The dataset provided by ISSDA consists of the genuine and unaltered data of 5,000 users over 536 days. These data have been anonymized to protect user identities. However, actual instances of energy theft cannot be identified due to customer privacy concerns. The solution to this lies in synthetically generated data by researchers.
Reviewer 4’ comments: Please revise some sentences, like "Figure 4 is explained as follows:".
Response: Thank you for the comment. All related sentences were revised.
Reviewer 4’ comments: The statement "SVM, LR, RF, KNN, DT algorithms were used with CNN for classification" lacks clarity. Are these classifiers essential for CNN classification? Additional details on the rationale behind combining these approaches are necessary.
Response: Thank you so much again for the valuable comments. Details are elaborated in Table 6. As seen in Table 6, most of the CNN-based classifiers used have achieved higher performance than native CNN. Additionally, all of them have obtained lower FPR. Additional details regarding the rationale behind combining these approaches have been added to the explaining of Table 6.
Finally and most importantly, the paper was checked again to improve its grammar and as well as removing the typographical errors in the revised version. In addition, all references cited in the article were revised, the spelling deficiencies were fixed, and it was automatically rewritten from the .bib file by using Zotero. Moreover, the article was completely revised and written in LaTeX format.
Thank you so much for your contributions.
Authors.
Round 2
Reviewer 1 Report
Comments and Suggestions for Authors
Thank you for the improvement. I am satisfied with the response from the authors.
Comments on the Quality of English LanguageThank you for the improvement. I am satisfied with the response from the authors.
Author Response
Authors’ Response to the Review Comments
Journal : MDPI, Sensors
Manuscript ID : Sensors-2822818
Title of Paper : Smart Grid Security: An Effective Hybrid CNN-based Approach for Detecting Energy Theft using Consumption Patterns
Authors : M.Zekeriya Gunduz, Resul Das
Date Sent : January 30, 2024
First of all, We would like to very much thankful to the Editor and Reviewers for their deep and thorough review. We have revised our existing research article in light of their useful suggestions and comments.
Responses to Comments of Reviewer 1
Reviewer 1’ comments: Thank you for the improvement. I am satisfied with the response from the authors.
Response: We would like to thank you very much for your valuable view and your comment that adds value to the article. As evaluated by the reviewer, we wanted to make a well-organized paper of our study about energy theft detection in the smart grid. So we are grateful for this evaluation.
Thank you so much for your contributions.
Authors.
Reviewer 2 Report
Comments and Suggestions for Authors
The authors have addressed all the issues; therefore, the manuscript is recommended for acceptance.
Author Response
Authors’ Response to the Review Comments
Journal : MDPI, Sensors
Manuscript ID : Sensors-2822818
Title of Paper : Smart Grid Security: An Effective Hybrid CNN-based Approach for Detecting Energy Theft using Consumption Patterns
Authors : M.Zekeriya Gunduz, Resul Das
Date Sent : January 30, 2024
First of all, We would like to very much thankful to the Editor and Reviewers for their deep and thorough review. We have revised our existing research article in light of their useful suggestions and comments.
Responses to Comments of Reviewer 2
Reviewer 2’ comments: The authors have addressed all the issues; therefore, the manuscript is recommended for acceptance.
Response: We would like to thank you very much for your valuable view and your comment that adds value to the article. As evaluated by the reviewer, we wanted to make a well-organized paper of our study about energy theft detection in the smart grid. So we are grateful for this evaluation.
Thank you so much for your contributions.
Authors.
Reviewer 4 Report
Comments and Suggestions for Authors
Thank you for incorporating the revisions based on the comments regarding the previous manuscript. It would be better if the following facts could also be addressed.
- Please reconsider rephrasing the sentence, "The minor and major contributions are as follows:". Could you clarify which contributions are considered minor and which are major?
- The contributions can also be summarized.
Author Response
Authors’ Response to the Review Comments
Journal : MDPI, Sensors
Manuscript ID : Sensors-2822818
Title of Paper : Smart Grid Security: An Effective Hybrid CNN-based Approach for Detecting Energy Theft using Consumption Patterns
Authors : M.Zekeriya Gunduz, Resul Das
Date Sent : January 30, 2024
First of all, We would like to very much thankful to the Editor and Reviewers for their deep and thorough review. We have revised our existing research article in light of their useful suggestions and comments. We hope that this minor revision has improved the paper to a level of your satisfaction.
Number wise answers to their specific comments /suggestions /queries are as follows.
Responses to Comments of Reviewer 4
Reviewer 4’ comments: Thank you for incorporating the revisions based on the comments regarding the previous manuscript. It would be better if the following facts could also be addressed.
Response: We would like to thank you very much for your valuable view and your comment that adds value to the article. As evaluated by the reviewer, we wanted to make a well-organized paper of our study about energy theft detection in the smart grid. So we are grateful for this evaluation.
Reviewer 4’ comments: Please reconsider rephrasing the sentence, "The minor and major contributions are as follows:". Could you clarify which contributions are considered minor and which are major? The contributions can also be summarized.
Response: Thanks for the valuable comment. To avoid confusion, the relevant sentence has been changed to "The main contributions of this paper can be summarized as follows:". Page 2 line 84.
Thank you so much for your contributions.
Authors.